# The Effect of *Lactobacillus plantarum* Extracellular Vesicles from Korean Women in Their 20s on Skin Aging

Chan Song Jo [1,†], Cheol Hwan Myung [1,†], Yeo Cho Yoon [2], Beom Hee Ahn [2], Jin Woo Min [3], Won Sang Seo [2,3], Dong Hwan Lee [4], Hee Cheol Kang [2,3], Yun Hoe Heo [5], Hyeong Choi [5], In Ki Hong [5] and Jae Sung Hwang [1,*,†]

1   Department of Genetic Engineering & Graduate School of Biotechnology, College of Life Sciences, Kyung Hee University, Yongin 17104, Gyeonggi-do, Korea; jchansong93@naver.com (C.S.J.); audjoin2@naver.com (C.H.M.)
2   Human & Microbiome Communicating Laboratory, GFC Co., Ltd., Hwasung 18471, Gyeonggi-do, Korea; yc.yoon@gfcos.co.kr (Y.C.Y.); bh.ahn@gfcos.co.kr (B.H.A.); seows@gfcos.co.kr (W.S.S.); michael@gfcos.co.kr (H.C.K.)
3   Green & Biome Customizing Laboratory, GFC Co., Ltd., Hwasung 18471, Gyeonggi-do, Korea; jw.min@gfcos.co.kr
4   Clinical Business Division, Korea Dermatology Research Institute, GFC Co., Ltd., Sungnam 13517, Gyeonggi-do, Korea; dh.lee@gfcos.co.kr
5   R&D Complex, HK Kolmar Co., Ltd., Seoul 30004, Korea; yhheo@kolmar.co.kr (Y.H.H.); jecniclous@kolmar.co.kr (H.C.); inkiaaa@kolmar.co.kr (I.K.H.)
*   Correspondence: jshwang@khu.ac.kr
†   These authors contributed equally to this work.

**Abstract:** Extracellular vesicles, which are highly conserved in most cells, contain biologically active substances. The vesicles and substances interact with cells and impact physiological mechanisms. The skin is the most external organ and is in direct contact with the external environment. Photoaging and skin damage are caused by extrinsic factors. The formation of wrinkles is a major indicator of skin aging and is caused by a decrease in collagen and hyaluronic acid. MMP-1 expression is also increased. Due to accruing damage, skin aging reduces the ability of the skin barrier, thereby lowering the skin's ability to contain water and increasing the amount of water loss. *L. plantarum* suppresses various harmful bacteria by secreting an antimicrobial substance. *L. plantarum* is also found in the skin, and research on the interactions between the bacteria and the skin is in progress. Although several studies have investigated *L. plantarum*, there are only a limited number of studies on extracellular vesicles (EV) derived from *L. plantarum*, especially in relation to skin aging. Herein, we isolated EVs that were secreted from *L. plantarum* of women in their 20s (*Lp*EVs). We then investigated the effect of *Lp*EVs on skin aging in CCD986sk. We showed that *Lp*EVs modulated the mRNA expression of ECM related genes in vitro. Furthermore, *Lp*EVs suppressed wrinkle formation and pigmentation in clinical trials. These results demonstrated that *Lp*EVs have a great effect on skin aging by regulating ECM related genes. In addition, our study offers important evidence on the depigmentation effect of *Lp*EVs.

**Keywords:** extracellular vesicles (EVs); exosome; skin aging; *Lactobacillus plantarum*; pigmentation

## 1. Introduction

Extracellular vesicles (EVs) are highly conserved lipid-membrane-enclosed vesicles found in most cells, including prokaryotes, eukaryotes, and archaea [1,2]. EVs contain a variety of biologically active substances such as proteins, lipids, nucleic acids, and metabolites. They reflect the state of the cells from the originating molecules, and communicate with neighboring or distant cells [3]. EVs include exosomes and micro-vesicles [4]. Exosomes are 30–200 nm vesicles secreted from multi-vesicular endosomes (MVEs), which are endosomes that make up numerous vesicles and undergo fusion with the plasma membrane [5]. Micro-vesicles are 100–1000 nm in size and are produced through budding with the plasma

membrane [6]. EVs derived from various cells interact with target cells and affect various physiological mechanisms, such as the immune response and inflammation [7].

Bacterial-derived EVs are classified into Gram-negative and Gram-positive bacteria-derived EVs. EVs from Gram-negative bacteria contain lipopolysaccharides (LPS) [8], and EVs from Gram-positive bacteria contain lipoteichoic acids (LTA) [9]. Bacterial-derived EVs are involved in transferring antibiotic resistance proteins to other bacteria [10] and induce communications between bacteria. They also function to eliminate competing bacteria by delivering a protein that degrades the peptidoglycans of competing bacteria [11]. Additionally, bacterial EVs can cause disease in the host [12]. For example, Staphylococcus aureus-derived EVs cause atopic dermatitis by delivering α-hemolysin to the skin [13]. Consequently, there are a number of research studies that use these characteristics and optimize the transporter mechanism to mediate drug delivery to specific cells or tissues [14–18].

*Lactobacillus plantarum* is a Gram-positive member of the genus Lactiplantibacillus, is rod-shaped and 3–8 μm in length [19], and produces lactic acid [20]. *L. plantarum* is found in many fermented products and has been associated with reducing allergic reactions as a probiotic, and lowering cholesterol and triglyceride levels [21–24]. In particular, *L. plantarum* suppresses various harmful Gram-positive and Gram-negative bacteria by secreting an antimicrobial substance from the human gastrointestinal tract [25,26]. *L. plantarum*, which is an aerotolerant Gram-positive bacteria, is also found on the skin, and research on the interactions with the skin are currently in progress [27–30]. However, the effect of *Lp*EVs on skin wrinkle formation not have been studied.

The skin is the largest organ in the human body and the most external-facing organ, in direct contact with the external environment [31]. Photoaging and skin damage occur because of extrinsic factors such as UV and external harmful factors; they also occur due to various inflammatory cytokines and intrinsic factors, such as ROS, that are generated during metabolism [32]. Wrinkles of the skin are a major indicator of skin aging [33]. The formation of wrinkles is caused by a decrease in the expression of collagen and hyaluronic acid, which are components of the extracellular matrix (ECM) [34]. Simultaneously, an increase in the expression of MMP-1, a metalloproteinase that degrades collagen, stimulates skin wrinkles [35]. In addition, aging of the skin reduces the ability of the skin barrier because of damage, thereby lowering the skin's ability to contain water, and increasing the amount of water loss [36].

Various bacteria and viruses reside in colonies in the stratum corneum and pores of the human skin. This coexistence is known as the microbiome [37]. The composition of the microbiome depends on the environment where the bacteria grow, a person's sex, and age [38,39]. This microbiome resides on the skin and maintains a balance with other surrounding communities, making it resistant when exposed to external pathogens, and it also serves a beneficial function in preventing infection in the body [40]. With aging, the composition of the microbiome changes due to exposure to UV and various chemicals. This change in the composition of the microbiome also accelerates skin aging, because the microbiome and the skin cells in the body interact with each other [41]. Substances such as proteins and lipids are secreted differently, due to various environment and stimuli, and are delivered through EVs [42].

This study confirmedthe anti-aging and anti-pigmentation effects of *Lp*EVs as demonstrated in in vitro test and clinical trials. We foundthat the number of *L. plantarum* bacteria in the skin of women in their 20s was higher than for women in their 50s, on average, and that the absence of *L. plantarum* was associated with skin aging. The results confirmed that *Lp*EVs, which were obtained from the skin of women in their 20s, improves skin aging such as skin wrinkling and elasticity. Therefore, these results indicate that the *Lp*EVs in young skin can be used as an effective anti-skin aging agent.

## 2. Materials and Methods

### 2.1. Isolation of Microorganisms

In this study, microorganisms were isolated from human skin and the following separation method was used: Gauze was rubbed on the women's skin suspensions, which were obtained by adding distilled water; then 150 μL of each sample were spread on de Man, Rogosa, and Sharpe (MRS) agar plates under aerobic condition at 37 °C. Single colonies were obtained and purified by transferring them to new MRS agar plates. The strain was kept in MRS broth medium that contained 30% glycerol at −70 °C.

### 2.2. 16S rRNA Gene Sequence and Phylogenetic Analysis

The genomic DNA isolation Kit (Gene all, Seoul, Korea) was used to isolate the genomic DNA of the strain, according to the manufacturer's instructions. The 16S ribosomal RNA (rRNA) gene was amplified from chromosomal DNA of a strain using the universal bacteria primer sets, and the full sequence was assembled with SeqMan software version 7.1 (DNASTAR Inc., Madison, WI, USA). The 16S rRNA gene sequence similarities between the strains and other related Lactobacillus species were obtained from the GenBank database. Multiple sequence alignments were performed using the CLUSTAL X program and calculated using the two-parameter Kimura method. Finally, a phylogenetic tree was constructed with the neighbor-joining and maximum-parsimony methods using the MEGA7 Program.

### 2.3. Microorganism Preparation

Single colonies of the microorganisms that belonged to the Lactobacillus plantarum species were inoculated and pre-incubated in MRS broth medium (KisanBio, Seoul, Korea) at 37 °C for 18 h. Next, the cultures' *Lp*EVs were rinsed three times with DPBS to remove residual medium, and incubated in 10% skim milk (BD, Franklin Lakes, NJ, USA) at 37 °C for 24 h.

### 2.4. Extracellular Vesicle Isolation

*L. plantarum*, which was extracted from human skin tissue for use in this study, was cultured in 10% skim milk and then Extracellular vesicles (EVs) were isolated. In detail, they were centrifuged at $4000 \times g$ for 10 min, and the EVs were purified with ultra-centrifugation (Hitachi, Chiyoda-ku, Tokyo, Japan) at $10,000 \times g$ for 30 min, and $150,000 \times g$ for 2.5 h. The EV-rich pellets were re-suspended at a final volume of 100 mL with distilled water (DW), and kept at 4 °C in a freezer, after filtering with a 0.22 μm bottle-top filter.

### 2.5. Nanoparticle Tracking Analysis (NTA)

Nanoparticle tracking analysis (NTA) was conducted with a Zetaview TWIN (Particle Metrix, Meerbusch, DE) to confirm the diameter and concentration of the extracellular vesicles. Lactobacillus species-derived extracellular vesicles (*Lp*EVs) were isolated from human skin, suspended in filtered DW at 20.15 °C and were irradiated with a blue-light laser wavelength ($\lambda = 488$ nm). The sample conductivity was performed at 42.19 μS/cm and the filter wavelength was measured with backscatter detection. Samples were measured with dilution (dilution factor was 500) on the equivalent sample aliquot. The data were analyzed using ZetaView Software (version 8.05).

### 2.6. Cell Culture

The human dermal fibroblasts (CCD986sk) were purchased from the American Type Culture Collection (Manassas, VA, USA) and incubated in DMEM high glucose medium (WelGene Inc., Daegu, Korea), supplemented with 10% fetal bovine serum (FBS, WelGene Inc., Daegu, Korea) and 1% penicillin/streptomycin (Hyclone Laboratories Inc., Logan, UT, USA) at 37 °C, in an atmosphere that contained 5% $CO_2$.

### 2.7. Cell Viability Assay

We investigated changes of viability in cells based on treatment with *Lp*EVs. The cell viability was determined using the cell proliferation reagent WST-1 (Dojindo Molecular Technologies Inc., Rockville, MD, USA). Cells were seeded in a 96-well plate at $1 \times 10^4$ cells/well in 200 μL of complete conditioned medium, supplemented with 10% FBS and 1% penicillin/streptomycin. Cells were incubated for 18 h at 37 °C in an atmosphere that contained 5% $CO_2$. Cells were then simultaneously treated with 0.625%, 1.25%, 5%, and 10% concentrations of dose-dependent *Lp*EVs, and incubated for 24 h at 37 °C and 5% $CO_2$. After incubation, 200 μL/well WST-1 reagents were added and incubated for 2 h at 37 °C. The cells were then measured absorbance against a background control with a microplate reader (BioTek Instruments, Inc., Winooski, VT, USA) at 450 nm.

### 2.8. LpEV Treatment Induces Elastase Inhibitory Activity

Elastase inhibitory activity was performed in Tris-HCL buffer (0.2 mM, pH 8.0). Porcine pancreatic elastase (Sigma-Aldrich, St. Louis, MO, USA) was dissolved to make a 5 mg/mL stock solution in distilled water (DW). As substrate, *N*-Succinyl-Ala-Ala-Ala-*p*-nitroanilide was dissolved in a buffer at 1.8 mM. The LpEVs were treated and incubated with the enzyme for 20 min before adding a substrate to begin the reaction. The final reaction mixture (total volume 200 μL) contained the buffer. distilled water (DW) was used as negative control. Elastase inhibitory activity was measured continuously for 30 min immediately afteradding the substrateusing a Microplate Reader (BioTek Instruments, Inc., Winooski, VT, USA) in 96-well micro-plates.

The percentage inhibition for elastase inhibitory activity is calculated by:

$$\text{Elastase inhibitory activity (\%)} = [(\text{OD}_{\text{control(DW)}} - \text{OD}_{\text{LpEV}})/\text{OD}_{\text{control(DW)}}] \times 100$$

### 2.9. mRNA Expression Analysis with Reverse Transcript PCR (RT-PCR)

We performed a RT-PCR analysis to investigate changes in mRNA expression, and to determine genes that were correlated with skin elasticity and treatment with *Lp*EVs. A TRIzol reagent (Sigma-Aldrich Chemical Co., St. Louis, MO, USA) was used to extract mRNA from CCD986sk dermal fibroblasts treated with *Lp*EVs. Assessment of the purity and integrity of the mRNA was performed using a Nano Drop™ 2000/2000c Spectrophotometer (Thermo Fisher scientific, Waltham, MA, USA) and analyzed at 260/280 nm. RT-PCR was conducted with primers for matrix metalloproteinase-1 (MMP-1), pro-collagen type I (COL1A1), and filaggrin (FLG). The primer sequences used in this study are shown in Table 1 and these primers, which were designed by ourselves, were used. The RNA template was reverse-transcribed using amfi-Rivert cDNA Synthesis Platinum Master Mix (GenDEPOT, Katy, TX, USA) and amplified by PCR using a C1000 Touch™ thermal cycler (Bio-rad, Hercules, CA, USA). The PCR program included an initial denaturation at 95 °C for 2 min, followed by 40 cycles of 30 s at 95 °C, 90 s at 62 °C, and 5 min at 70 °C. In this study, all mRNA expression experiments were repeated more than three times.

**Table 1.** Each primer sequences and Tm information used in this study.

| Primer | | Primer Sequence | Tm (°C) |
|---|---|---|---|
| Actin | Forward | 5′—CATGAAGTGTGACGTGGACA—3′ | 58 °C |
| | Reverse | 5′—CAGGGCAGTGATCTCCTTCT—3′ | |
| COL1A1 | Forward | 5′—GACCTCAAGATGTGCCACTC—3′ | 58 °C |
| | Reverse | 5′—CCAGTCTCCATGTTGCAGAA—3′ | |
| MMP-1 | Forward | 5′—CCCAGCGACTCTAGAAACAC—3′ | 58 °C |
| | Reverse | 5′—GCCTCCCATCATTCTTCAGG—3′ | |
| Filaggrin | Forward | 5′—GCTGAAGGAACTTCTGGAAAAG—3′ | 62 °C |
| | Reverse | 5′—GCCAACTTGAATACCATCAGAAG—3 | |

### 2.10. Protein Expression Analysis with Western Blot

We investigated the effects of Hyaluronidase 2 (HAS2), which is known to lyase hyaluronic acid in vitro, to confirm the skin moisturizing effects of *Lp*EV treatments. Dermal fibroblasts were stimulated for 24 h and harvested. The proteins in cells were extracted with 1 × RIPA buffer and a proteinase/phosphate inhibitor buffer, and run on 10% SDS-PAGE gels. Proteins were then blotted on the PVDF membrane, and immune-detected with primary antibodies against HAS2 (ab140671, abcam, Chambridge, UK), and with a secondary anti-mouse antibody (ab6728, abcam, Chambridge, UK). A ChemiDoc imaging system was used for detection (ChemiDoc XRS+, Bio-Rad, Hercules, CA, USA). The HAS2 protein volume was normalized by actin protein expression.

### 2.11. Preparation of Skin Application Solutions

Mannitol 5% that contained Lactobacillus extracellular vesicles was used for an experimental group, and Mannitol 5% was used as a control group. Thereafter, phosphate-buffered saline was added and adjusted to 100%. Samples were stored at 5 °C to 25 °C.

### 2.12. Volunteer Recruitment and Selection

The test period was from 26 November 2020 to 24 December 2020. Of a total of 20 volunteers, 16 Korean women were tested and 4 dropped out (IRB Number: KDRI-IRB-20936). The criteria for selecting volunteers were as follows: (1) a person who voluntarily wrote and signed an informed consent form after the principal investigator, or a person delegated by the principal investigator, fully explained and informed the research subject; (2) a healthy person without acute or chronic physical diseases, including skin diseases; (3) a subject that could complete a follow-up during the testing period. Treated methods are directly used by study subjects. They mix agent 1 and 2, evenly did on face twice a day in the morning and evening. And then, they measured skin condition every 2 weeks for 4 weeks. Before the test, the volunteers first removed any waste or debris; then, after resting at 20–24 °C with 40–60% RH, for 30 min, they took pictures using MARK Vu and F-ray equipment. Thereafter, the measurement site was partitioned and instrumental measurement was performed. Photography and device evaluation were performed in the same manner after 2-week and 4-week periods.

### 2.13. Skin Contour Measurement

F-Ray and Moire techniques were used to measure skin contours. The elasticity was measured by shooting at a total of seven angles (front, left and right 30°, 45°, 60). An 18-megapixel camera was used.

### 2.14. Skin Image Measurement

A MARK Vu (PSI PLUS, Suwon-si, Korea) was used for skin image measurement. A continuous light source that used four types of LEDs—general light, polarized light, ultraviolet light, and glossy light—was used. Using the device's Detail Logic program, we were able to assess 13 different skin conditions such as pores, wrinkles, blemishes, and sebum, from high-resolution photos.

### 2.15. Skin Wrinkles, Elasticity, and Dermal Density Measurements

Skin wrinkles were measured using ANTERA 3D (Miravex, Dublin, Ireland) which is a high-resolution three-dimensional image measuring device that obtained three-dimensional images of the skin by using an optical method and a mathematical algorithm. The image measured the depth and width of fine wrinkles, the roughness of the skin, and the number of pores. A Cutometer® MPA 580 (Courage & Khazaka, Cologne, Germany) was used for skin elasticity analysis. Dermal density was measured using Ultrasound (DermaLab Skin, CORTEX TECHNOLOGY, Hadsund, Denmark) equipment.

### 2.16. Statistical Analysis

Significance was confirmed using the Minitab 19 (Minitab® 19.2, Minitab Inc., State College, PA, USA) program. The paired *t*-test was used to compare the values measured before and after the test, and the significance was confirmed at the level of $p < 0.05$, $p < 0.01$, $p < 0.001$, through repeated measure ANOVA, by repeating measurements three or more times.

## 3. Results

### 3.1. S rRNA and Phylogenetic Analysis of Lactobacillus plantarum

We previously found that the population of *L. plantarum* was significantly higher in the skin of women in their 20s than those in their 50s, on average (Figure 1). We hypothesized that the decrease in *L. plantarum* might be related to skin aging. To confirm this hypothesis, we collected specimens from the foreheads of women in their 20s, and *L. plantarum* was isolated on MRS agar plates. In order to identify whether the isolated strain was actually *L. plantarum*, we extracted genomic DNA and amplified 16S RNA chromosomal DNA. The amplified sequence was assembled using SeqMan software and a BioEdit program. The results were consistent with *L. plantarum* in the GenBank database (Figure 2), and this strain was used for subsequent experiments.

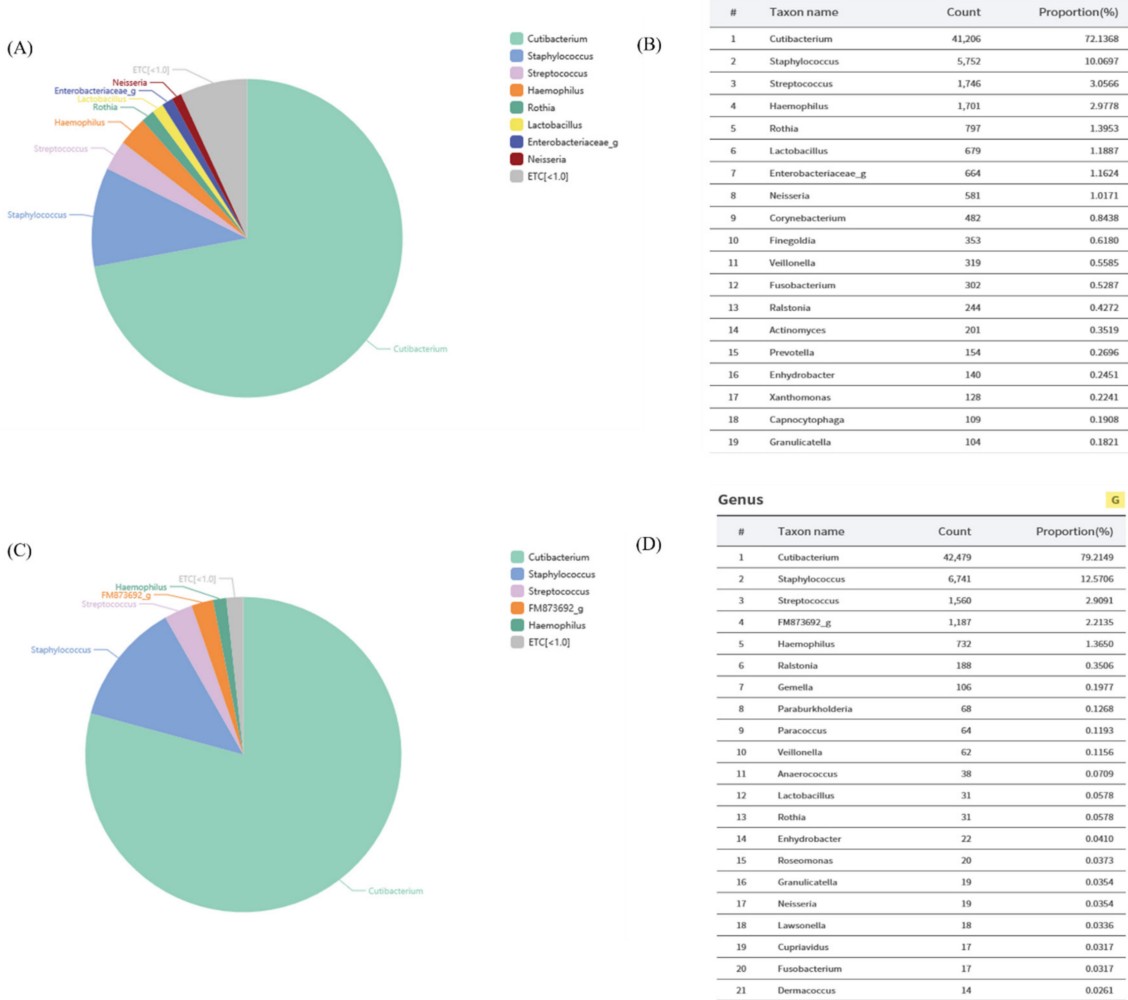

**Figure 1.** Quantitative data of *L. plantarum* ratio of skin of women in their (**A**,**B**) 20 s and (**C**,**D**) 50 s.

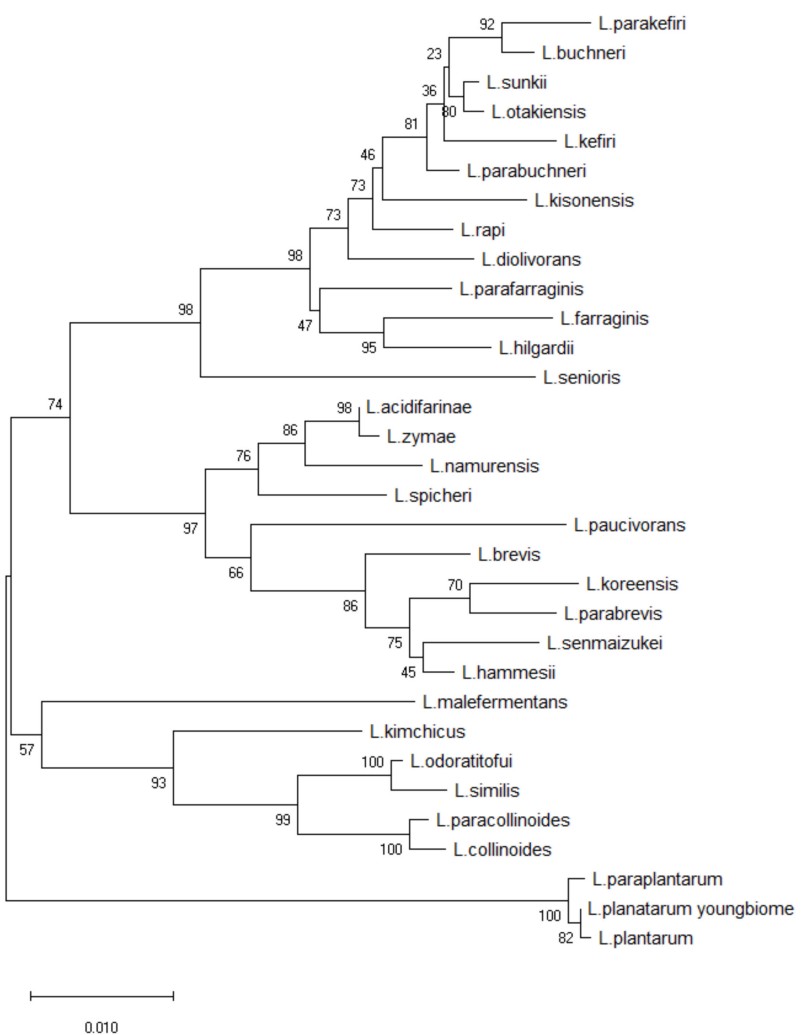

**Figure 2.** Phylogenetic tree based on 16S rRNA gene sequences of *Lactobacillus plantarum* isolated from skin of women in their 20′s.

### 3.2. Lactobacillus plantarum Actively Secretes EVs

We isolated extracellular vesicles secreted from *L. plantarum* of women in their 20s using ultracentrifugation, analyzed *Lp*EVs with a nanoparticle tracking analysis (NTA) video, and confirmed their size and distribution (Figure 3A). NTA analysis revealed that the *Lp*EVs exist in exosome forms (Figure 3B). The EV particles had an average diameter of $126.5 \pm 56.4$ nm, an average size of 50–200 nm, and the concentration of EVs was 35.86 μg/mL (per mL). The number of particles per mL of EVs and mg of protein concentration was $9.1 \times 10^9$ per 1 mL, and $2.53 \times 10^{11}$ per 1 mg of protein (Figure 3C). From these results, we determined that *L. plantarum* isolated from the skin of women in their 20s actively secretes EVs, with sizes that range from 50 to 200 nm.

### 3.3. LpEV Treatment Induces Cell Proliferation and Regulates ECM Degradation-Associated Gene Expression

Senescent cells are characterized by their inability to proliferate [43]. Therefore, we validated proliferation to confirm the effect of *Lp*EVs in fibroblasts. The result means that the *Lp*EVs have an effect on proliferation in fibroblasts at concentrations of 2.5 and 5%, but not 10% (Figure 4A). We then investigated ECM-related gene expression. The cells that make up the dermis are surrounded by an extracellular matrix (ECM) that connects them and allows the cells to maintain their shape. First, we irradiated UVA and then treated *Lp*EVs in fibroblasts.We examined MMP-1 expression and the amount of elastase to confirm

the effect of *Lp*EVs on ECM degradation (Figure 4B). The mRNA level of MMP-1, an enzyme that degrades the ECM, decreased significantly based on the assessment of 0.625% of *Lp*EVs. In addition, in order to confirm the skin elasticity effect of *Lp*EVs treatment, fibroblasts were treated with *Lp*EVs, and then the elastase activity was analyzed. As a result, we found that *Lp*EVs increased the inhibition of elastase activity, which is a peptidase, from approximately 20% at a concentration of 1.25% to 40% at a 10% concentration (Figure 4C). These results showed that the EVs of women in their 20s affect cell proliferation, and can inhibit the expression or activities of ECM degradation enzymes and peptidase.

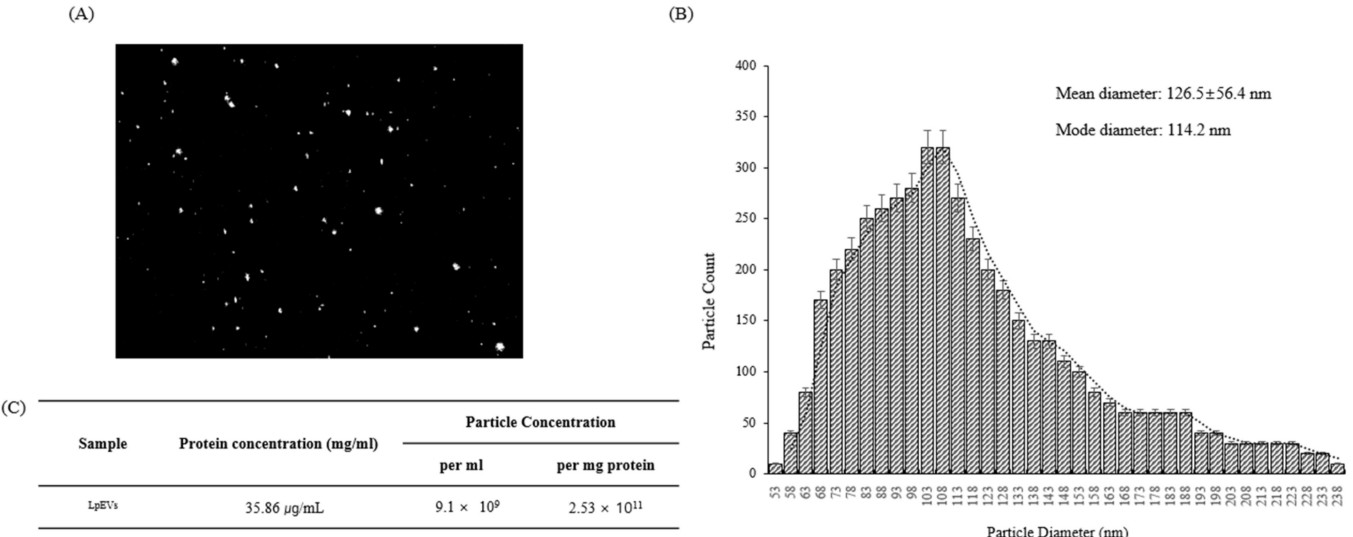

**Figure 3.** Purification and characteristics of *Lactobacillus plantarum*-derived extracellular vesicles (*Lp*EVs): (**A**) a representative frame from one of the *Lp*EVs' nanoparticle tracking analysis videos. The purified EVs were diluted 1:500 in distilled water; (**B**) the particle size and number of *Lp*EVs determined by nanoparticle tracking analysis (NTA); (**C**) protein and particle concentration of *Lp*EVs.

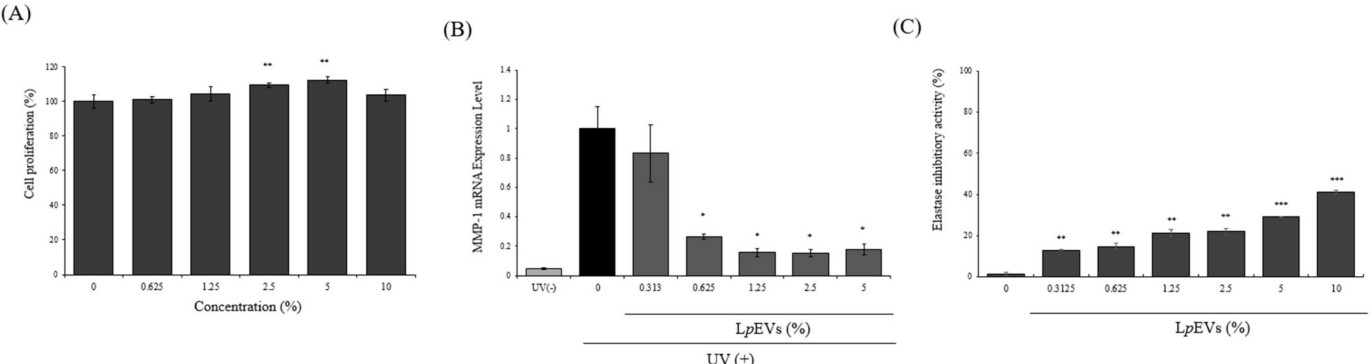

**Figure 4.** Evaluation of the effects of *Lp*EVs on MMP-1 mRNA expression levels using RT-PCR in CCD986sk: (**A**) cell Proliferation of *Lp*EVs in CCD986sk dermal fibroblasts. Cell proliferation assays were performed on cells treated with *Lp*EVs in a dose dependent manner (** $p < 0.01$); (**B**) the MMP-1 expression levels in CCD986sk after irradiation of UVA and treatment of *Lp*EVs (* $p < 0.05$); (**C**) the elastase inhibitory activity was measured in a dose dependent manner in CCD986sk (** $p < 0.01$, *** $p < 0.001$).

### 3.4. LpEV Treatment Induces ECM Production-Associated Gene Expression

We examined mRNA expression of Type 1 procollagen following *Lp*EVs treatment in CCD986sk dermal fibroblasts, to investigate influence of *Lp*EVs on ECM production as well as ECM degradation. The mRNA level of Type 1 procollagen increased by a factor of 1.7 times compared with TGF-beta that was used as a positive control. *Lp*EVs increased

in a dose-dependent manner and to about 1.6 times the baseline measurement at 10% concentration (Figure 5A). Filaggrin is involved in epidermal homeostasis and maintains the skin barrier function. Filaggrin expression is generally reduced as aging occurs [44]. Accordingly, we investigated whether *Lp*EVs affect the expression of filaggrin in CCD986sk dermal fibroblasts. As a result, *Lp*EVs increased the mRNA expression of filaggrin by more than 2 times at a concentration of 2.5–10% (Figure 5B). In addition, the protein expression of HAS2, which induces hyaluronic acid synthesis and plays a crucial role as a crosslinking of ECM [45], decreased slightly at low concentrations of EV (0.625–2.5%), but increased by about 20–30% at 5% and 10% concentrations (Figure 5C). Therefore, *Lp*EVs increase the expression of ECM components and the enzymes related to ECM.

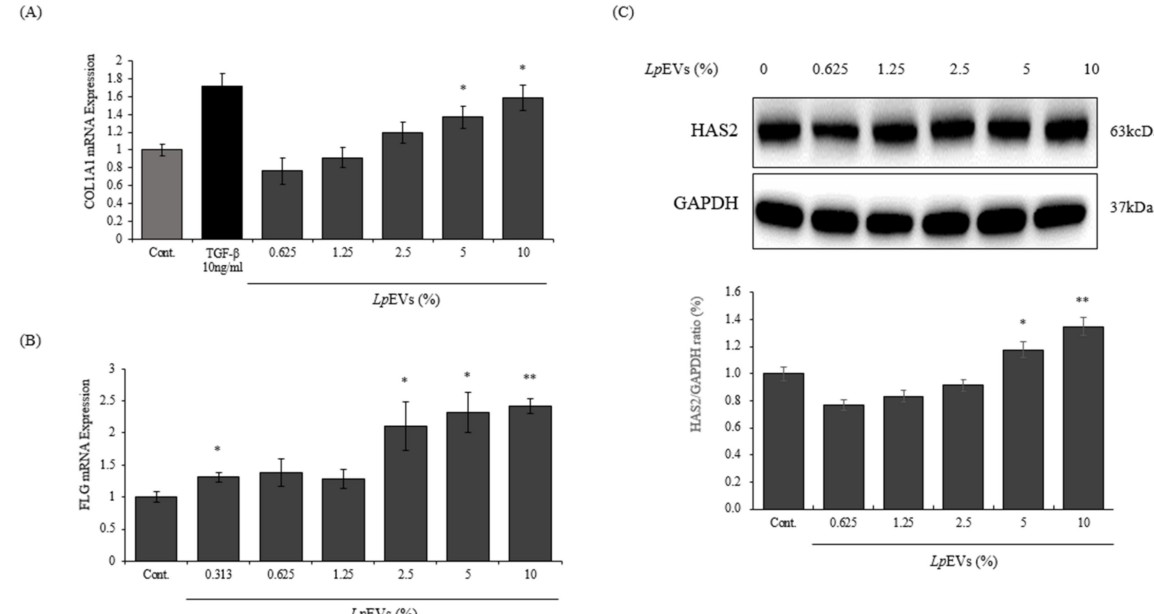

**Figure 5.** Evaluation of the effects of *Lp*EVs on collagen, filaggrin, and HAS2 expression levels in CCD986sk: (**A**) collagen mRNA expression levels (\* $p < 0.05$); (**B**) filaggrin mRNA levels (\* $p < 0.05$, \*\* $p < 0.01$); (**C**) HAS2 protein levels in CCD986sk after 24h of treatment (\* $p < 0.05$, \*\* $p < 0.01$).

### 3.5. LpEV Treatment Suppresses Wrinkle Formation in Clinical Trials

Ultraviolet radiation or damage caused by aging reduce skin elasticity and cause wrinkle formation [46,47]. We conducted clinical assessments on 16 skin wrinkles around the eyes of women in their 50s, on average, to determine whether *Lp*EVs can restore the aging index in aging skin (Table 2). We applied *Lp*EVs (or placebo EVs) to the wrinkles around the eye, and the wrinkles were measured at 0, 2, and 4 weeks with the Antera 3D. The indentation index value (A.U.), which determines the degree of wrinkles around the eyes, decreased by 8.9% at 2 weeks and 15.89% at 4 weeks compared to week 0, whereas there was no change in the placebo group (Figure 6A). Skin elasticity improved by 14.76% at 2 weeks and by 27.07% at 4 weeks (Figure 6B). The Antera 3D image showed that treatment of *Lp*EVs gradually decreased the distribution and formation of wrinkles at 2 and 4 weeks. In contrast, it was difficult to confirm a significant wrinkle change in the placebo group (Figure 6C). These results suggest that *Lp*EV treatment improves skin elasticity and suppresses wrinkle formation.

**Table 2.** Subject information of clinical trials.

| Subjects of Clinical Trials (IRB Number: KDRI-IRB-20936) * | | | |
|---|---|---|---|
| **Gender** | **Age** | | **Average Age** |
| Female | 40's | 50's | Age 50 |
| | n = 6 | n = 10 | |
| | Total n = 16 | | |

*: Each data partner obtained the necessary Institutional Review Board (IRB) approval or exemption.

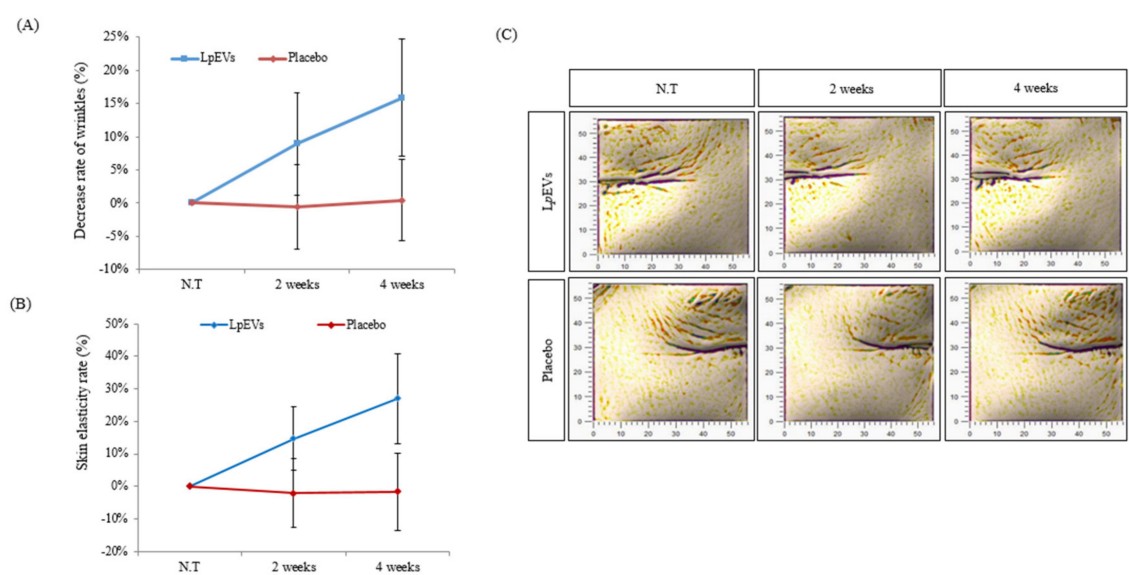

**Figure 6.** Evaluation of the effects of *Lp*EVs on wrinkle formation: (**A**) eye-wrinkle improvement assessments in clinical trials; (**B**) epidermidis elasticity improvement results in clinical trials; and (**C**) Antera 3D image (Wrinkle: Small) from clinical trials.

### 3.6. LpEV Treatment Moisturizes Skin and Enhances Skin Density

A major characteristic of skin aging is the change in moisture content and skin density. The moisture content of the skin decreases as the skin barrier weakens, and skin density is also reduced due to a decrease in the ECM [45,48,49]. Therefore, we tried to assess whether *Lp*EVs can affect and improve the water content and density in the skin. Unlike the placebo group, which had no significant effect, the *Lp*EVs increased water content by 10.79% at 2 weeks and 21.40% at 4 weeks (Figure 7A). We confirmed that skin density increased at the 2-week and 4-week assessments in both groups, using ultrasound (Figure 7B). Image quantification images showed an increase in skin density, but the increased rate of the density (39.30%) of the *Lp*EV group was higher than that of the placebo (15.19%) (Figure 7C). Therefore, we confirmed that *Lp*EVs suppress the reduction in skin moisture content and increase skin density.

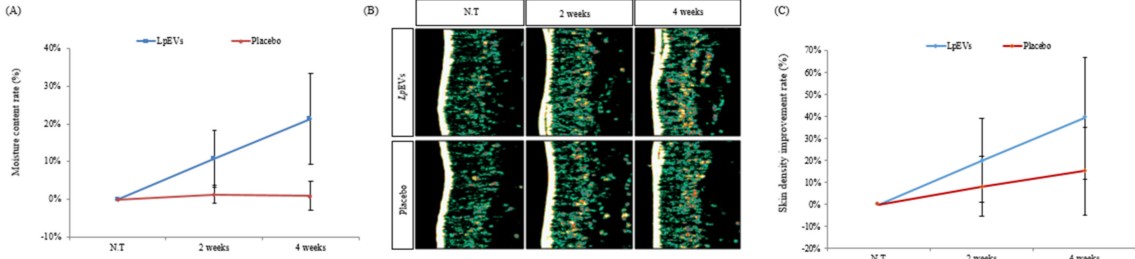

**Figure 7.** Evaluation of the effects of *Lp*EVs on moisture contents and skin density: (**A**) skin moisture improving effects; (**B**) ultrasound images; and (**C**) a numerical graph of the skin density improvement rate.

*3.7. LpEV Treatment Suppresses Skin Pigmentation Caused by Aging*

Another factor of skin aging is skin pigmentation [50]. We investigated whether the *Lp*EVs had a whitening effect that suppressed pigmentation caused by aging. As a result, for patients treated with *Lp*EVs, unlike the placebo, the pigmentation of the lesion sites was decreased at the 2-week and 4-week assessments (Figure 8A,C). In the *Lp*EV treatment group, skin density improved by 3.87% at 2 weeks and 8.7% at 4 weeks (Figure 8B). Therefore, these data showed that *Lp*EVs have an effect on pigmentation caused by skin aging. Consequently, *Lp*EVs have a great anti-aging effect.

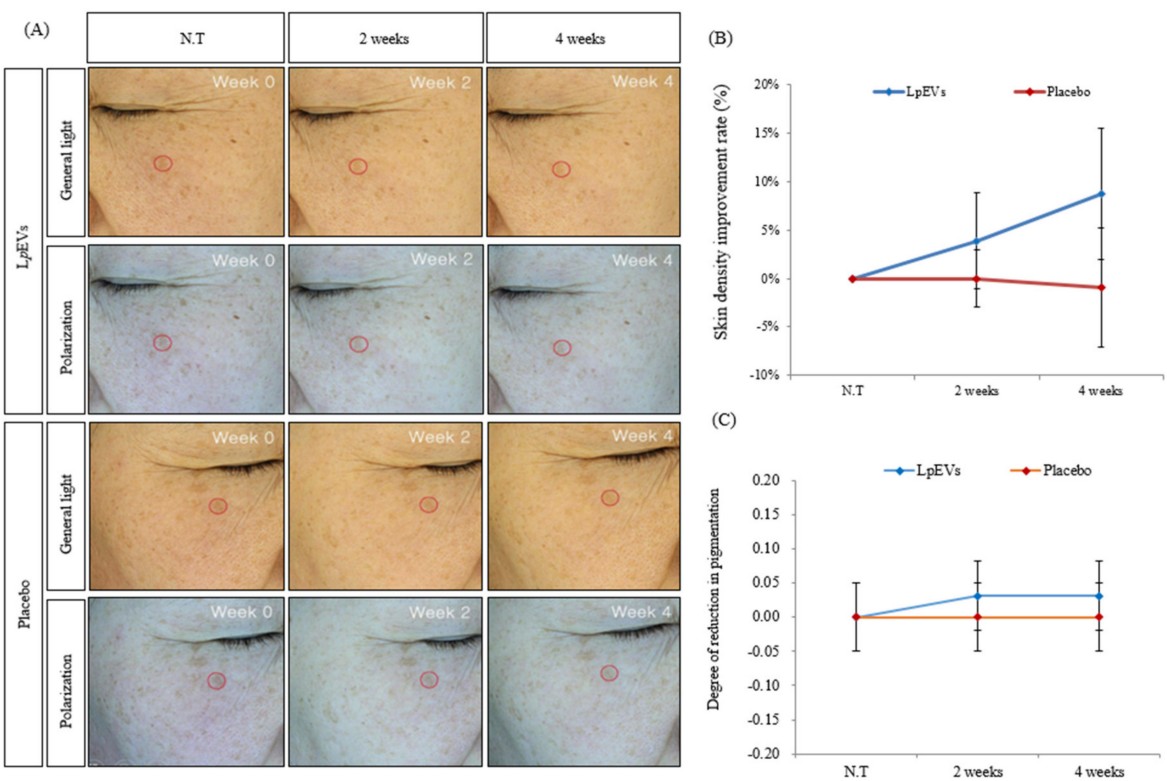

**Figure 8.** Evaluation of the effects of pigmentation reduction through image analysis (Mark Vu) with *Lp*EVs in clinical trials: (**A**) Mark Vu image; (**B**) the numerical graph of the skin density improvement rate of (**A**); (**C**) degree of reduction in pigmentation in visual reading.

## 4. Discussion

Skin aging is caused by external factors such as UV rays and internal factors, which include telomere shortening [51–53]. Recently, studies on the skin microbiome have attracted considerable interest, because it has been identified as a factor that can impact skin aging [54,55]. The difference in the microbiome composition of young and aged skin

suggests that the microbiome may be involved in skin aging [56]. The microbiome has direct contact with the outermost skin, and also interacts with the skin cells by secreting extracellular vesicles (EVs), such as exosomes, that contain biologically active molecules [4]. Therefore, we hypothesized that differences in the microbiome between women in their 20s and 50s, on average, would be related to skin aging.

*Lp*EVs have an effect on the cell proliferation of CCD986sk dermal fibroblasts (Figure 4A). It is known that many EVs have an anti-aging effect, and can increase skin density by restoring or increasing the proliferation of fibroblasts [57]. Likewise, the data showed that the EVs of *L. plantarum* increased cell proliferation. We then investigated the *Lp*EVs in this experiment-induced processes that inhibited ECM degradation (Figure 4B,C) and increased the expression of proteins related to ECM such as collagen, filaggrin and HAS2 (Figure 5). Based on our results, we suggest that *L. plantarum* could be applied to help prevent skin aging.

We conducted clinical assessments on women that were in their 50s women, on average, and confirmed the aging index, which is caused by a decrease in skin elasticity and wrinkle formation. We found that *Lp*EVs could reduce wrinkle formation (Figure 6). The loss of moisture content in the skin arises from damage to the skin barrier [48]. Interestingly, *Lp*EVs increased the moisture content of the skin (Figure 7). However, future studies are required to assess whether the *Lp*EVs restore the skin barrier or the moisture content is increased by the ECM improvements, such as collagen and hyaluronic acid. In addition, another characteristic caused by damage to the skin barrier is an increase in the amount of water loss in the skin. Therefore, it is important to also assess the amount of moisture loss in the skin.

We also determined that *Lp*EVs suppressed pigmentation caused by aging skin for women in their 50s, on average (Figure 8). Our results showed that the *Lp*EVs can influence aging-induced pigmentation. Further studies on the depigmentation effect can further elucidate the applications for *Lp*EVs.

In this study, we demonstrated that women in their 20s had a higher population of *L. plantarum* in their skin microbiome than women in their 50s, on average. Additionally, *Lp*EVs could suppress aging factors (Figure 9). Consequently, the results suggest that *Lp*EVs, which are components of the skin microbiome, can be applied as an effective anti-aging agent to improve skin aging, and also as an effective anti-pigmentation agent.

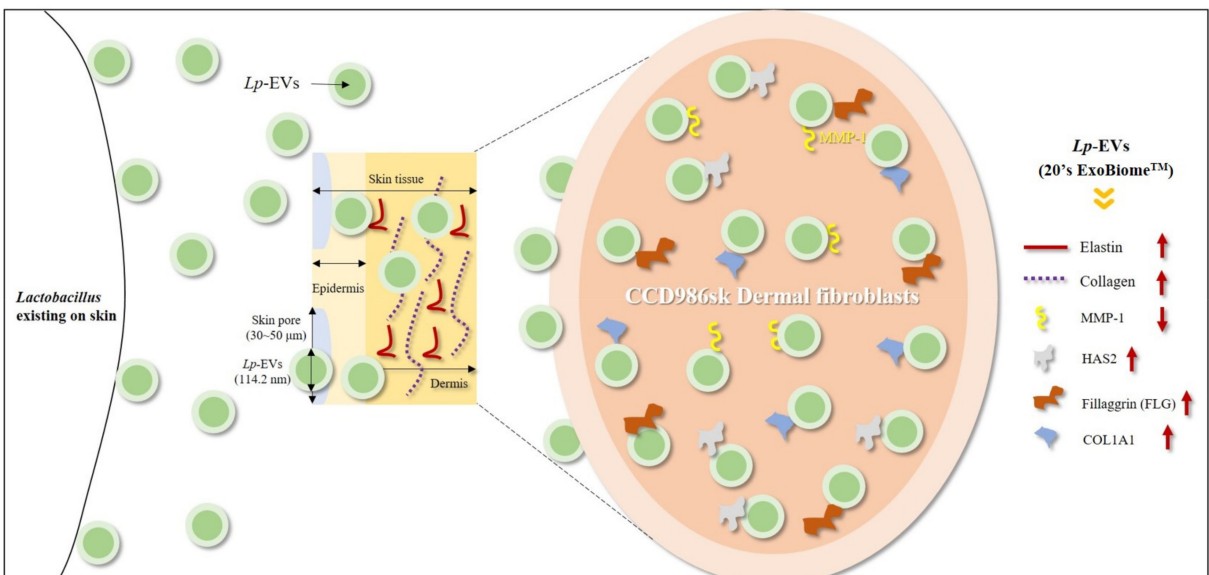

**Figure 9.** The anti-aging effects of extracellular vesicles derived from *Lactobacillus plantarum* isolated from the skin of women in their 20s.

**Author Contributions:** Conceptualization, J.S.H., C.S.J., C.H.M. and Y.C.Y.; methodology, C.S.J., H.C.K., Y.C.Y. and C.H.M.; software, B.H.A., D.H.L., Y.H.H.; formal analysis, J.W.M., W.S.S., H.C.K.; investigation, C.S.J., C.H.M., I.K.H. and J.S.H.; Writing—Original Draft, C.S.J. and H.C.; Writing— Review & Editing, C.S.J., C.H.M. and J.S.H.; supervision, J.S.H.; funding acquisition, J.S.H. All authors have read and agreed to the published version of the manuscript.

**Funding:** This research received no external funding.

**Institutional Review Board Statement:** The study was conducted according to the guidelines of the Declaration of Helsinki, and approved by the Institutional Review Board of Korea Dermatology Research Institute (protocol code KDRI-IRB-20936 and date of approval: 12 January 2021).

**Informed Consent Statement:** Informed consent was obtained from all subjects involved in the study.

**Conflicts of Interest:** The authors declare no conflict of interest.

## Abbreviations

| | |
|---|---|
| EV | extracellular vesicle |
| ECM | extracellular matrix |
| *Lp*EVs | EVs that were secreted from *L. plantarum* of women in their 20s |
| MMP-1 | matrix metalloproteinase-1 |
| COL1A1 | pro-collagen type I |
| FLG | filaggrin |

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
