# Peer review of "The Effect of Lactobacillus plantarum Extracellular Vesicles from Korean Women in Their 20s on Skin Aging"

_cimb, doi:10.3390/cimb44020036_

Round 1

Reviewer 1 Report

For the Ms-cimb-149751 some minor revisions are required.
- The authors for their investigation have chosen the CCD986sk fibroblast cell line, to make the observations complete, in particular the effect of LpEVs would be suitable for a cell line of keratinocytes or better reconstructed skin models.
- Is the real composition of the LpEVs known?
- The sequence of the primers used is not present in the text.
- Other information on the primers used would be useful, TA, n.cycles, source from which the authors designed the sequences.
- Western blot for HAS2 and GAPDH seems artificially retouched.
- There are numerous typos, the text needs to be double-checked carefully

Reviewer 2 Report

The manuscript reports the results of a study aimed to verify the effects of Lactobacillus plantarum extracellular vesicles on skin aging.  Extracellular vesicles from L. plantarum isolated of Korean women in their 20s were evaluated for their in vitro effects on fibroblasts (cell proliferation, mRNA expression for matrix metalloproteinase-1, type I pro-collagen and filaggrin, elastase activity, expression of hyaluronidase 2) as well as for the effect on wrinkle formation in a clinical trial.
Although the results are of interest, I suggest to re-write the manuscript improving also the language: there are many inaccuracies in the text, and often it is not clear what has been done and which are the observed effects. 
Reading the Materials and Methods section, sometimes if not clear or it is not reported how the studies were conducted. 
For instance, at line 121, the sentence “Extracellular vesicles were isolated from whey and cultured in 10% skim-milk of LpEVs that had been extracted from human skin tissue” does not specify how the vesicles have been isolated.
Other examples: At lines 150-152, authors write “The cells were then measured against a background control with a microplate reader (BioTek 151 Instruments, Inc., Winooski, VT, USA) at 450 nm”. They should specify that they measured absorbance at 450 nm.
Lines 155-157:  “A trizol reagent (Sigma-Aldrich Chemical Co., St. Louis, USA) was used to extract mRNA from LpEVs treated with CCD986sk dermal fibroblasts”. Authors should write that fibroblast have been exposed to LpEVs and not vice versa.
Subsection 2.11. Volunteer recruitment and selection: Authors should more clearly report the scheme of treatment.
Results section:
Lines 251-253: Authors should better comment the effect on proliferation (no concentration-dependent effect was observed).
Lines 257-258 and Figure 4B: From Figure 4B, it seems that fibroblasts have been exposed to UV radiation but this is not specified in the text.  
Lines 258-262: The evaluation of elastase activity has not been reported in Materials and Methods section. It is not clear if the effect has been recorded on the pure enzyme or in fibroblasts.

Author Response

The manuscript reports the results of a study aimed to verify the effects of Lactobacillus plantarum extracellular vesicles on skin aging.  Extracellular vesicles from L. plantarum isolated of Korean women in their 20s were evaluated for their in vitro effects on fibroblasts (cell proliferation, mRNA expression for matrix metalloproteinase-1, type I pro-collagen and filaggrin, elastase activity, expression of hyaluronidase 2) as well as for the effect on wrinkle formation in a clinical trial.
Although the results are of interest, I suggest to re-write the manuscript improving also the language: there are many inaccuracies in the text, and often it is not clear what has been done and which are the observed effects. 
Reading the Materials and Methods section, sometimes if not clear or it is not reported how the studies were conducted. 

We attached the certificate of editing
For instance, at line 121, the sentence “Extracellular vesicles were isolated from whey and cultured in 10% skim-milk of LpEVs that had been extracted from human skin tissue” does not specify how the vesicles have been isolated.

I corrected it.

  1. plantarum, extraction from human skin tissue, used to this study, was cultured on 10% skim-milk and then isolated Extracellular vesicles (EVs) from supernatant including whey except to curd layer.

Other examples: At lines 150-152, authors write “The cells were then measured against a background control with a microplate reader (BioTek 151 Instruments, Inc., Winooski, VT, USA) at 450 nm”. They should specify that they measured absorbance at 450 nm.

I corrected it

The cells were then measured absorbance against a background control with a microplate reader (BioTek Instruments, Inc., Winooski, VT, USA) at 450 nm.

Lines 155-157:  “A trizol reagent (Sigma-Aldrich Chemical Co., St. Louis, USA) was used to extract mRNA from LpEVs treated with CCD986sk dermal fibroblasts”. Authors should write that fibroblast have been exposed to LpEVs and not vice versa.

I corrected it.

A trizol reagent (Sigma-Aldrich Chemical Co., St. Louis, USA) was used to extract mRNA from CCD986sk dermal fibroblasts treated with LpEVs.

Subsection 2.11. Volunteer recruitment and selection: Authors should more clearly report the scheme of treatment.

We corrected it

Treated methods is that as study subject's directly own used, after mix agent 1 and 2, evenly did on face twice a day in the morning and evening. And then, visited in the center every 2 weeks for 4 weeks and measured skin condition.

Results section:
Lines 251-253: Authors should better comment the effect on proliferation (no concentration-dependent effect was observed).

The result means that the LpEVs have an effect on proliferation in fibroblasts at concentration of 2.5 and 5%, not 10% (Figure 4A).

Lines 257-258 and Figure 4B: From Figure 4B, it seems that fibroblasts have been exposed to UV radiation but this is not specified in the text.  

We corrected it

First, we were irradiated UVA on cells and then treated LpEVs. We examined MMP-1 expression and the amount of elastase to confirm the effect of LpEVs on ECM degradation (Figure 4B).

(B) The MMP-1 expression levels in CCD986sk after irradiation of UVA and treatment of LpEVs (*p < 0.05).

Lines 258-262: The evaluation of elastase activity has not been reported in Materials and Methods section. It is not clear if the effect has been recorded on the pure enzyme or in fibroblasts.

We corrected it.

2.8. LpEV treatment induces elastase inhibitory activity

Elastase inhibitory activity was performed in Tris-HCL buffer (0.2 mM, pH 8.0). Porcine pancreatic elastase (Sigma-aldrich, MO, USA) was dissolved to make a 5 mg/mL stock solution in distilled water (DW). As substrate, N-Succinyl-Ala-Ala-Ala-p-nitroanilide was dissolved in buffer at 1.8 mM. The LpEV were treated and incubated with the enzyme for 20 minutes before adding substrate to begin the reaction. The final reaction mixture (total volume 200 μl) contained buffer. Negative controls were used distilled water (DW). Elastase inhibitory activity was measured immediately adding of the substrate and then continuously for 30 minutes using a Microplate Reader (BioTek Instruments, Inc., Winooski, VT, USA) in 96 well micro-plates.

The percentage inhibition for elastase inhibitory activity is calculated by:

Elastase inhibitory activity(%)=[(ODcontrol(DW) - ODLpEV) / ODcontrol(DW)] x 100

Round 2

Reviewer 2 Report

I read the revised manuscript. Considering that authors improved the manuscript modifying some parts according to my previous comments, the manuscript can be accepted for publication in CIMB. However, even though the authors attached the certificate of editing, I suggest to improve it from a linguistic point of view.  

Author Response

We revised manuscript again, so please see the attachment. 
